# A Prospective Comparative Study of Mastication Predominance and Masticatory Performance in Kennedy Class I Patients

**DOI:** 10.3390/healthcare9060660

**Published:** 2021-06-01

**Authors:** Kohei Kinoshita, Yoichiro Ogino, Kyosuke Oki, Yo Yamasaki, Yoshihiro Tsukiyama, Yasunori Ayukawa, Kiyoshi Koyano

**Affiliations:** 1Section of Implant and Rehabilitative Dentistry, Division of Oral Rehabilitation, Faculty of Dental Science, Kyushu University, Fukuoka 812-8582, Japan; koheikinoshita@dent.kyushu-u.ac.jp (K.K.); yamayo@dent.kyushu-u.ac.jp (Y.Y.); ayukawa@dent.kyushu-u.ac.jp (Y.A.); 2Section of Fixed Prosthodontics, Division of Oral Rehabilitation, Faculty of Dental Science, Kyushu University, Fukuoka 812-8582, Japan; 3Section of Dental Education, Division of Oral Biological Sciences, Faculty of Dental Science, Kyushu University, Fukuoka 812-8582, Japan; tsuki@dent.kyushu-u.ac.jp; 4Division of Advanced Dental Devices and Therapeutics, Faculty of Dental Science, Kyushu University, Fukuoka 812-8582, Japan; koyano@dent.kyushu-u.ac.jp

**Keywords:** mastication predominance, masticatory performance, Kennedy class I, removable partial dentures, posterior occlusal support

## Abstract

Mastication predominance in Kennedy class I (KC I) patients has not been well defined. This study aimed to investigate mastication predominance and masticatory performance in KC I patients, including the significance of remaining posterior teeth and removable partial-denture (RPD) treatment. KC I patients who had differences in the number of posterior teeth between left and right sides (D+) and KC I patients who had no differences (D−) were enrolled. Healthy dentate (HD) subjects were also registered as a positive control. Mastication predominance, defined by mastication predominance index (MPI; range 0–100%) calculated from electromyogram activities during voluntary chewing, and masticatory performance were evaluated at pre- and post-RPD treatment. Pre-MPI in KC I D+ was significantly higher than in HD. RPD treatment could significantly improve MPI and masticatory performance in both KC I groups. However, there were significant differences in masticatory performance between each KC I group and HD, regardless of RPD treatment. It was considered that the mastication predominance in KC I patients was affected by the difference in the number of remaining posterior teeth. RPD treatment could improve mastication predominance and masticatory performance in KC I patients, although the latter was not similar to HD group.

## 1. Introduction

It has been shown that even healthy individuals chew more on either the left or right side of the dental arch [1,2,3,4,5,6]. Mastication predominance is defined as the habit of chewing predominantly on one side and has been reported to be the cause of temporomandibular disorders, temporomandibular joint disc displacement, and asymmetrical loss of teeth [4,5,6,7,8,9,10]. Excessive mastication predominance might be related to laterality in stomatognathic functions, such as jaw movement patterns, bite force, and masticatory performance [11,12,13]. Although substantial evidence is lacking, it is reasonable to consider that bilateral chewing is generally recommended to prevent these conditions.

Some previous studies indicated that the individuals who lost several posterior teeth, also known as shortened dental arch (SDA), showed no significant difference in the scores of oral-health-related quality of life compared to the patients with conventional removable partial dentures (RPDs) [14,15]. However, in our previous study, we found that patients with unilateral and bilateral missing posterior teeth chewed on one side more predominantly than healthy dentate subjects, and there was a significant difference in masticatory performance between healthy dentate subjects and patients with bilateral missing posterior teeth [16,17]. These findings suggest the significance of prosthetic treatment for SDA patients to some extent.

Distal extension RPDs are generally used for individuals with bilateral missing posterior teeth to improve masticatory function. There have been many studies that reported the improvement in oral functions following prosthetic treatment [18,19,20]. Our previous study clearly showed that RPD treatment for Kennedy class II (KC II) patients, who lost unilateral posterior teeth, significantly improved mastication predominance [21]. This study suggested that KC II patients predominantly chewed on the one side with more teeth (healthy side) compared to the other side (unilaterally posterior edentulous side). This finding implied the significance of the number of remaining teeth in mastication predominance. However, detailed data of mastication predominance and masticatory function in patients with bilateral missing posterior teeth (Kennedy class I patients; KC I patients) have not been reported, and the significance of remaining teeth, especially the number of posterior teeth, has not been evaluated. 

The purpose of the present study was to investigate mastication predominance and masticatory performance in KC I patients from the viewpoint of the number of remaining posterior teeth and RPD treatment. The null hypotheses tested in the present study werethat RPD treatment would not improve mastication predominance and masticatory performance, and there would be no significant effect of the difference in the number of remaining teeth between left and right sides on mastication predominance and masticatory performance in KC I patients.

## 2. Materials and Methods

### 2.1. Study Population

The subjects in this study were the patients who visited the Department of Prosthodontics, Kyushu University Hospital, between October 2018 and July 2020. The following patients were included as the subjects: patients over 20 years old with missing posterior teeth bilaterally (KC I) in one jaw and with complete dental arch with natural teeth or fixed prosthetic devices in opposite jaw; and patients who would be scheduled to undergo bilateral RPD treatment. The following individuals were excluded from HD and KC I groups: those receiving continuous dental treatment except for RPD treatment and periodontal maintenance; those exhibiting systemic illness or dental disease that might affect mastication; those with jaw dysfunction and/or pain, such as temporomandibular disorders; and those with compromised mental capacity due to dementia or other psychiatric diseases. All participants provided the written informed consents that were approved by Kyushu University Institutional Review Board for Clinical Research (approval no. 2019-167). The patients with KC I were classified into two groups: one with differences between the number of posterior teeth (molars and premolars) on the left and right side (KC I D+), and one without the differences (KC I D−). In addition, healthy dentate (HD) volunteers who had a complete dental arch were enrolled from the staffs of Kyushu University as a positive control group because we could not enroll a sufficient number of HD subjects from the patients whose ages were similar to the KC I patients.

### 2.2. Maximum Occlusal Force (MOF) 

It has been reported that MOF could play a crucial role in masticatory performance [22,23,24]. To evaluate a masticatory-performance-related factor, MOF was compared between KC I patients. MOF was measured using the occlusal-force-analyzing system, as similar to the previous studies [22,23,24]. A pressure-sensitive sheet (Dental Prescale Ⅱ, GC, Tokyo, Japan) was placed on the dental arch and was pressed with maximum clenching force for 3 s in the intercuspal position. The sheet was analyzed using a measuring device (Bite force analyzer, GC, Tokyo, Japan) to calculate MOF.

### 2.3. Test Food

Test food (gummy jelly: Glucolumn; GC, Tokyo, Japan) was used to assess mastication predominance and masticatory performance. The size of test food was Ø15 × 8 mm/piece. These gummy jellies were used to measure masticatory performance in the previous studies [22,23,24,25].

### 2.4. Objective Evaluation of Mastication Predominance 

Mastication predominance was evaluated in our previous studies [16,17,21,26]. Electromyogram (EMG) activities during voluntary chewing in both the left and right masseter muscles were recorded using a portable EMG recording unit (ProComp Infiniti; Thought Technology, Montreal, Canada) and disposable Ag/AgCl surface electrodes (T3402M—Triode™ electrode; Thought Technology). The sampling frequency for EMG signals was 2048 Hz. Bipolar electrodes, with an inter-electrode distance of 20 mm, were set on the middle of the masseter muscle parallel to the orientation of muscle fibers after cleaning the skin surface with ethanol. The subjects were instructed to maintain their physiological rest position with the test food on their tongue prior to commencement of mastication. They were asked to chew the test food freely and swallow it. Since the reliability of this method had been confirmed in a previous study [26], this measurement procedure was conducted once. After this session, subjects were instructed to chew gummy jelly for 10 strokes on the right and left sides (designated chewing), respectively. Subsequently, subjects were instructed to perform maximum voluntary clenching for 3 s three times. These recordings were used as a reference for determining the preferred chewing side. The EMG data were saved directly to a personal computer. The above procedure was performed twice; the first measurement was performed before treatment, and the second was performed at least one month after treatment. Raw EMG signals were converted to root mean square (RMS). RMS waveforms during maximum voluntary clenching (MVC) were set as 100% MVC, and %MVC for 100% MVC of both left and right sides in each stroke were calculated. The side with greater values was determined to be the mastication side. Mastication frequency on the left and right sides during free mastication was recorded, and the following formula was used to determine mastication predominance:{mastication predominance value = [(number of right-side chewing strokes − number of left-side chewing strokes)/(number of right-side chewing strokes + number of left-side chewing strokes)] × 100 (%)}

The absolute value (%) was then set as the mastication predominance index (MPI). An MPI of 0% indicated that mastication was conducted evenly on the left and right sides, whereas an MPI of 100% indicated mastication was conducted only on either the left or right side. In addition, MPIs before and after RPD treatment were defined as pre- and post-MPI, respectively.

### 2.5. Objective Evaluation of Masticatory Performance 

To assess the effect of RPD treatment on oral rehabilitation, masticatory performance was measured as the previous studies evaluated [22,23,24,25]. The participants were instructed to chew the same gummy jelly (Glucolumn; GC, Tokyo, Japan) voluntarily for 20 s. Crushed gummy jelly was moved to a cup with saliva and rinsing water, and the concentration of glucose dissolved in water was measured using a measuring device (Glucosensor GS-1, GC, Tokyo, Japan). The measurement was performed three times, and the average value was calculated. The values of masticatory performance before and after RPD treatment were defined as pre- and post-masticatory performance, respectively.

### 2.6. Statistical Analysis

Numerical data were presented as median and interquartile range (IQR). Profiles of the subjects (age, the period from delivery of RPD to second (post) measurement, number of posterior occlusal supports, and maximum occlusal force) were statistically compared between KC I D+ and KC I D− using Mann–Whitney U test. In addition, effect size was calculated to assess the strength of association between the variables [27]. To analyze the data, following statistical comparisons were conducted. 

Evaluation of the difference of the subject profiles: differences in age, the period required for second measurement after RPD delivery, the number of posterior occlusal supports, and MOF between KC I D+ and KC I D− (Mann–Whitney U test)Comparison of the initial MPI among HD, KC I D+, and D− without RPD (before RPD treatment: pre-MPI): differences among HD-MPI, pre-MPI in KC I D+, and KC I D− (Kruskal–Wallis with multiple comparison)Comparison of the initial masticatory performance among HD, KC I D+, and D− without RPD (before RPD treatment: pre-masticatory performance): differences among HD-masticatory performance, pre-masticatory performance in KC I D+, and KC I D− (Kruskal–Wallis with multiple comparison)Evaluation of the effect of RPD treatment on MPI: differences between pre-MPI and post-MPI in KC I D+ and KC I D−, respectively (Wilcoxon signed-rank test)Evaluation of the effect of RPD treatment on masticatory performance: difference between pre- and post-masticatory performance in KC I D+ and KC I D−, respectively (Wilcoxon signed-rank test)Comparison of MPI among HD, KC I D+, and D− with RPD (after RPD treatment: post-MPI): difference in among HD-MPI, post-MPI in KC I D+, and KC I D− (Kruskal–Wallis with multiple comparison)Comparison of masticatory performance among HD, KC I D+, and D− with RPD (after RPD treatment: post-masticatory performance): difference in post-masticatory performance among HD-masticatory performance, post-masticatory performance in KC I D+, and KC I D− (Kruskal–Wallis with multiple comparison)

A value of *p* < 0.05 was considered statistically significant. All statistical analyses were performed using IBM SPSS Statistics 19 (IBM, Chicago, IL, USA).

## 3. Results

### 3.1. Subjects

The patients included in this study were 44 KC I patients (D+: 22 patients, D−: 22 patients, respectively) and 20 HD volunteers. Profiles of KC I patients are shown in Table 1. There were no significant differences between KC I D+ and KC I D− in all items (*p* > 0.05, Mann–Whitney U test). Above all, there was no difference in MOF between KC I D+ and KC I D−, suggesting MOF, as one of mastication performance-related factors, was statistically similar. Table 2 shows the posterior teeth distribution in KC I patients. The number of HD group was 20 (11 males and 9 females, median age: 27.5 years old IQR: 26.75–29.25).

### 3.2. Comparisons of MPI and Masticatory Performance among the Patients in Pre-KC I (D+ and D−) and HD

The results of pre-MPI of patients with KC I (D+ and D−) and HD-MPI were shown in Figure 1. A wide range of pre-MPI values was observed, especially in KC I D+. The statistical analyses revealed that pre-MPI in KC I D+ was significantly higher than HD-MPI (*p* < 0.05, Kruskal–Wallis with multiple comparison), and no significant differences were detected between KC I D+ and KC I D− and KC I D− and HD.

The results of pre-masticatory performance of patients with KC I (D+ and D−) and HD-masticatory performance are shown in Figure 2. The values of pre-masticatory performance in KC I D+ and D− were significantly lower than HD-masticatory performance (*p* < 0.001, Kruskal–Wallis with multiple comparison). These results suggested that masticatory performance in KC I patients without RPDs was significantly lower than HD.

### 3.3. Comparisons of MPI and Masticatory Performance in Patients with KC I (D+ and D−) between Pre- and Post-RPD Treatment

To evaluate the influence of RPD treatment on mastication predominance and masticatory performance, pre- and post-measurement values were statistically compared. The results of statistical comparison between pre- and post-MPI in each group (KC I D+ and KC I D−) are shown in Figure 3. A wide range of MPI values was also observed in all groups. In KC I D+ and KC I D−, RPD treatment could decrease MPI significantly (*p* < 0.05, Wilcoxon signed-rank test). Furthermore, Figure 4 clearly indicates that RPD treatment could significantly enhance masticatory performance in KC I D+ and KC I D−, respectively (*p* < 0.05, Wilcoxon signed-rank test). This finding suggests that RPD treatment was performed correctly from the aspect of oral rehabilitation.

### 3.4. Comparisons of MPI and Masticatory Performance between the Patients in Post-KC I and HD

Finally, post-MPI and post-masticatory performance were statistically analyzed among HD, KC I D+, and KC I D−. There were no significant differences in MPI among three groups (*p* > 0.05, Kruskal–Wallis with multiple comparison) (Figure 5). However, the values of post-masticatory performance in KC I D+ and KC I D− were significantly lower in comparison with HD-masticatory performance, despite the significant improvments of masticatory performance by RPD treatment in both KC I groups (*p* < 0.001, Kruskal–Wallis with multiple comparison) (Figure 6)

## 4. Discussion

Mastication predominance refers to the preferred chewing side [1,2,3,4,5,6]. Our previous studies confirmed validity of MPI to assess mastication predominance and evaluated mastication predominance in healthy dentate subjects, partially edentulous patients and Kennedy class Ⅱ (KC Ⅱ) patients [16,17,21,26]. These studies also indicated that KC Ⅱ patients tended to masticate predominantly on the healthy side with more teeth. The results were attributed to the distribution of remaining teeth, which means that KC Ⅱ patients lost their teeth unilaterally, and it is reasonable to consider that they tend to masticate on the healthy side, which means the side with more teeth. Furthermore, prosthetic interventions with RPDs or implant prostheses to restore the unilateral posterior edentulous area were effective to improve mastication predominance (MPI) [21]. However, there have been no detailed studies that demonstrated the characteristics of mastication predominance in KC I patients. We focused on the number of remaining posterior teeth and the difference in the number of teeth between the left and right side in KC Ⅱ patients and aimed to investigate mastication predominance. In addition, the effect of RPD treatment for KC I patients on mastication predominance and masticatory performance was also evaluated.

At first, we would like to discuss the calculation method of MPI and test food. As described above, this method was developed by our group and was already presented in several previous papers [16,17,21,26]. We believe that the evaluation using this method is reliable and already established. In the previous studies, several test foods were used. In the present study, gummy jelly, which has been widely used for the evaluation of masticatory performance, was adopted for the test food. This gummy jelly was used for the assessment of masticatory performance and was approved by Ministry of Health, Labour, and Welfare of Japan. We believe that there are no concerns about using this gummy jelly. However, other test foods that were used in the previous studies have specific features during mastication [16,17,21,26]. The evaluation using one test food might be a limitation of this study, unfortunately.

The number of the subjects might be insufficient because all subjects were enrolled from the patients whom the authors treated during the study period. However, we calculated effect size to evaluate the strength of association in addition to statistical comparisons [27]. All profile items of KC I D+ and D− patients in this study were not significantly different (Table 1). Especially, the number of posterior occlusal supports and MOF were known to contribute to mastication and masticatory performance [22,23,24]. These analyses presented that both KC I patients could be almost similar and be appropriate for statistical comparisons.

Our results clearly presented that KC I D+ patients showed significantly higher pre-MPI than HD subjects. In our previous study, KC I patients showed higher MPI compared to HD subjects [17]. Our previous studies also demonstrated that KC Ⅱ patients, who had the differences in the number of remaining teeth between the left and right side, showed higher MPI than HD subjects [16,21]. These findings, including the result of this study, suggested that patients tended to chew on the side with more teeth. Regarding masticatory performance, KC I D+ and D− patients showed significantly lower masticatory performance than HD subjects. Some studies showed that partially edentulous patients, such as KC I and Ⅱ patients, known as SDA patients, presented lower masticatory function [22,23,24,28], although oral-health-related quality of life was almost similar [14,15]. These studies might support our results.

RPD treatment for KC I patients could improve MPI significantly in KC I D+ and KC I D− patients. These improvements resulted in no significant differences among HD, KC I D+, and KC I D− groups. These findings clearly indicate that RPD treatment could play a critical role in the improvement of mastication predominance. Similar findings were observed in our previous study, which evaluated the effect of prosthetic interventions on MPI improvement in KC Ⅱ patients [21]. RPD treatment could increase the number of posterior occlusal supports and could contribute to the reduction of MPI even in KC I D− patients, who had same number of posterior teeth on both left and right sides. The finding in KC I D− might reveal that the increase in the number of posterior occlusal supports by RPD was effective in the improvement of MPI. However, we need to emphasize that the difference of the characteristics between KC I and Ⅱ must be considered, and the wide range of MPI must be recognized, which might mean that the impact of RPD treatment might be different among individuals.

Although the values of post-masticatory performance in both KC I groups were significantly improved compared to pre-masticatory performance as described above, our analyses revealed significantly lower post-masticatory performance in both KC I groups compared to the values in HD group. Various factors contribute to masticatory performance, such as age, occlusal force, and the number of remaining teeth and occlusal supports [22,23,24]. In this study, HD subjects were younger than KC I patients and, obviously, had more remaining teeth and occlusal support. It is reasonable to consider that masticatory performance at the same level of younger HD subjects was hardly achieved in older KC I patients, even though they were rehabilitated with RPDs.

Mastication predominance might be observed in most of the population to some extent. Dental factors, such as occlusion, cusp form, contacts in lateral movements, and the number of posterior teeth, might influence mastication predominance [12,29]. On the other hand, mastication predominance might be more subject to central control [13,30,31]. It has been commonly accepted that excessive mastication predominance has a high, potentially traumatic effect on dentition, jaw muscles, and temporomandibular joint. The previous studies suggested that long-term mastication predominance might cause asymmetry of facies [32,33]. Partially edentulous patients might need prosthetic interventions to prevent these disorders and to improve mastication predominance and masticatory performance, according to the suggestions of the present study.

## 5. Conclusions

Within the limitation of this study, the difference in the number of remaining teeth between the left and right sides might be related to mastication predominance in KC I patients in the absence of RPD, because KC I D+ patients showed higher MPI than HD subjects but not KC I D−. Prosthetic intervention (RPD treatment) can improve mastication predominance and masticatory performance in KC I patients, whereas further studies will be required using other methods and to assess the effect of prosthetic interventions, such as dental implants, on these functions.

## Figures and Tables

**Figure 1 healthcare-09-00660-f001:**
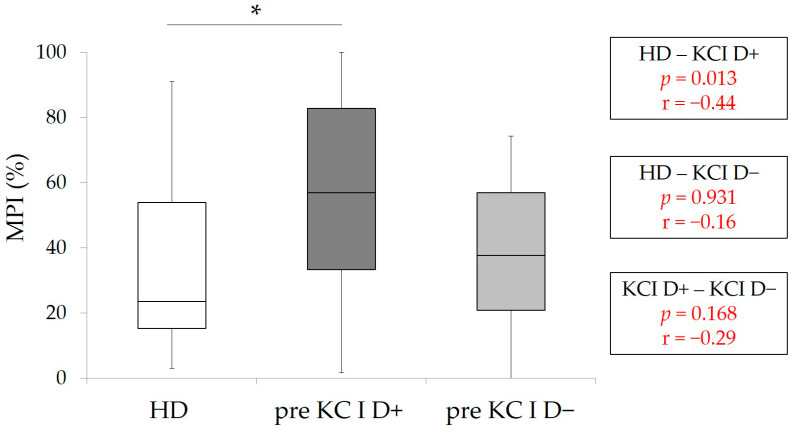
MPI in HD (control group), pre-MPI in KC I D+ and D−. * *p* < 0.05, Kruskal–Wallis with multiple comparison, r: effect size.

**Figure 2 healthcare-09-00660-f002:**
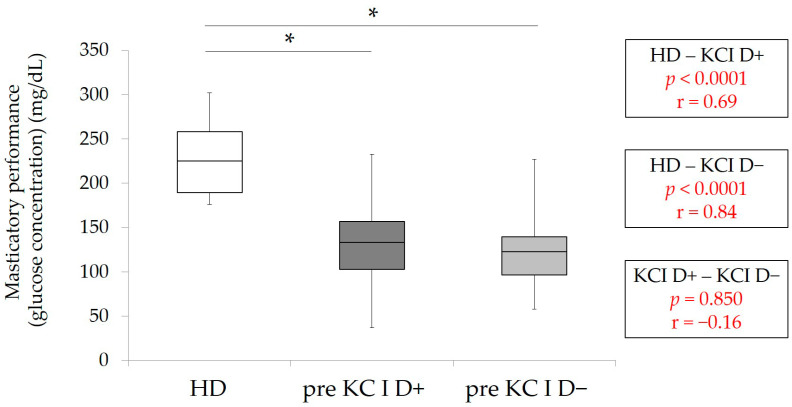
Masticatory function in HD (control group), pre-masticatory function in KC I D+ and D−. * *p* < 0.001, Kruskal Wallis with multiple comparison, r: effect size.

**Figure 3 healthcare-09-00660-f003:**
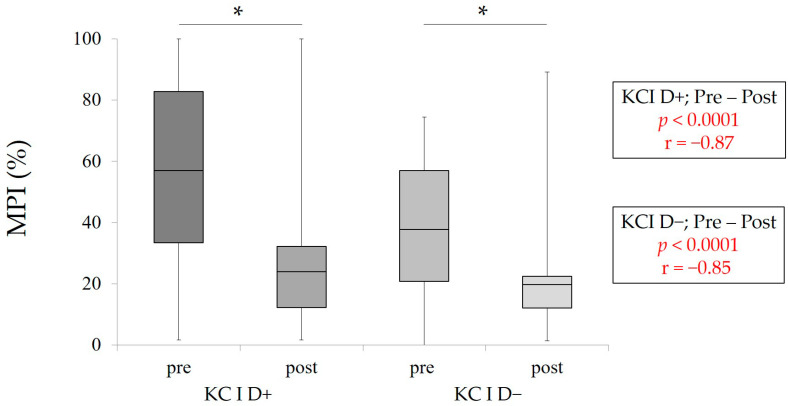
Pre- and post-MPI in in KC I D+ and D−. The statistical analyses were conducted between pre- and post-values in KC I D+ and D−, respectively * *p* < 0.05, Wilcoxon signed-rank test, r: effect size.

**Figure 4 healthcare-09-00660-f004:**
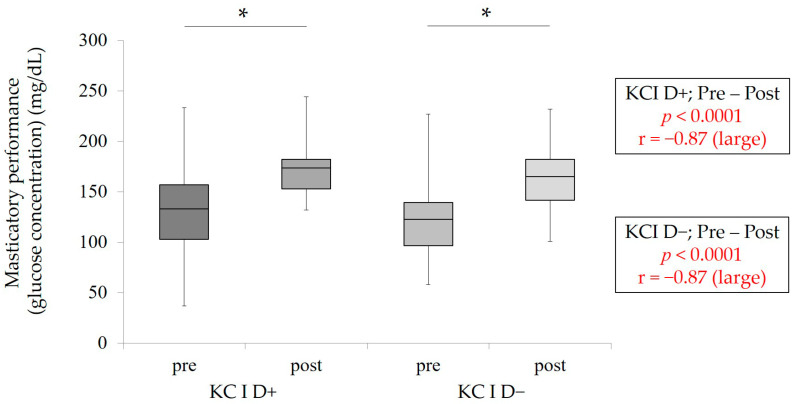
Pre- and post-masticatory function in KC I D+ and D−. The statistical analyses were conducted between pre- and post-values in KC I D+ and D−, respectively * *p* < 0.05, Wilcoxon signed-rank test, r: effect size.

**Figure 5 healthcare-09-00660-f005:**
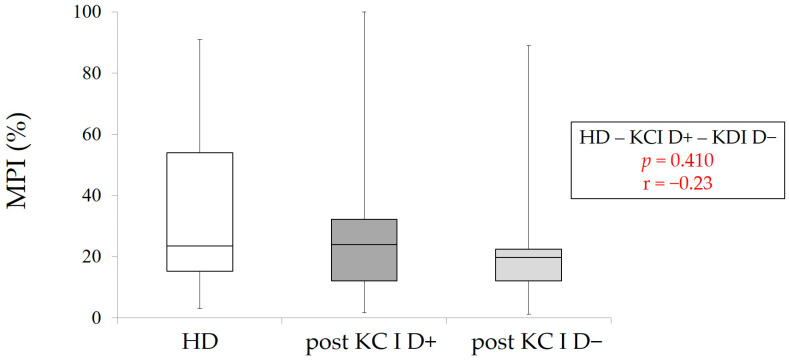
MPI in HD (control group), post-MPI in KC I D+ and D−. *p* > 0.05, Kruskal–Wallis with multiple comparison, r: effect size.

**Figure 6 healthcare-09-00660-f006:**
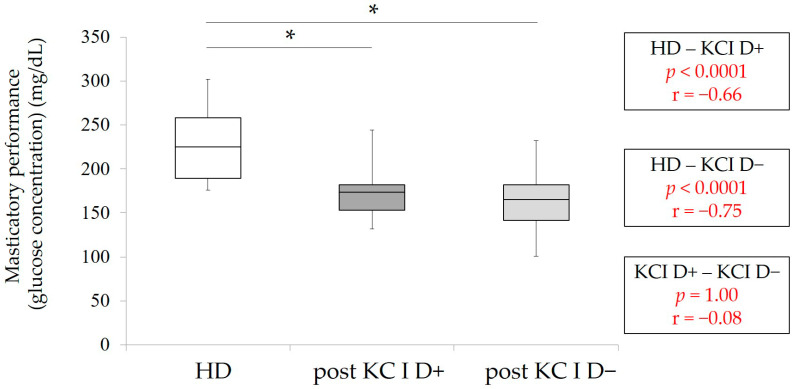
Masticatory function in HD (control group), post-masticatory function in KC I D+ and D−. * *p* < 0.001, Kruskal–Wallis with multiple comparison, r: effect size.

**Table 1 healthcare-09-00660-t001:** Summary of patients’ profiles (median and IQR). No statistical differences between KC I D+ and D− (*p* > 0.05, Mann–Whitney U test).

	KC I *n* = 44	KC I D+ *n* = 22	KC I D− *n* = 22	*p*-Value	Effect Size (r)
Age	74 (70–77.5)	75 (71–78)	73 (67.5–77.25)	0.95	0.09
Gender (male:female	14:30	4:18	10:12	0.55	−0.29
The period from delivery of RPD to post measurement (month)	6 (3–6)	5 (3–6)	6 (4–6)	0.757	−0.47
Number of posterior occlusal support	2 (1–3)	2.5 (1.3–3)	2 (0–4)	0.358	−0.17
Maximum occlusal force (*N*)	389.7 (193.8–500.3)	401.75 (340.2–561.3)	372.1 (147.95–516.5)	0.752	0.13

**Table 2 healthcare-09-00660-t002:** Posterior teeth distribution in KC I D+ and D− patients.

Group	Number of Posterior Occlusal Supports (One Side + the Other Side)	Number (Premolar)	Number (Molar)	Number (Patients)	Gender (Male & Female)	Age
D+	1 (1 + 0)	1	0	6	2:4	78 (76.25–79)
	2 (2 + 0)	2	0	5	0:5	76 (69–77)
	3 (3 + 0)	2	1	2	1:1	76 (75.5–76.5)
	3 (2 + 1)	3	0	7	1:6	72 (63–77)
	4 (3 + 1)	3	1	1	0:1	72
	5 (3 + 2)	4	1	1	0:1	74
D−	0 (0 + 0)	0	0	9	2:7	72 (68–77)
	2 (1 + 1)	2	0	7	2:5	74.5 (71.5–77.5)
	4 (2 + 2)	4	0	5	5:0	75 (66.25–77)
	6 (3 + 3)	4	2	1	1:0	64

## Data Availability

The datasets used and/or analyzed during the current study are available from the corresponding author on reasonable request.

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
