# Peer review of "A Prospective Comparative Study of Mastication Predominance and Masticatory Performance in Kennedy Class I Patients"

_healthcare, 2021, doi:10.3390/healthcare9060660_

Round 1

Reviewer 1 Report

Summary of manuscript: This manuscript reports the investigation of the mastication predominance and masticatory performance in KC I patients including the significance of remaining posterior teeth and removable partial denture (RPD) treatment. This report concludes that the difference in the number of remaining teeth between left and right sides might be related to mastication predominance in KC I patients in the absence of RPD because KC I D+ patients showed higher MPI than HD subjects, but not KC I D-.

Overall manuscript

  • P2 L61 Authors should describe the difference of number of posterior teeth in KC I, not KC I D+ and D-.
  • Authors should discuss the difference between pre– and post–MPI in KC I D- in Figure 3. If it was the same number of posterior teeth, it would not be no significant differences between pre-and post-MPI in KC I D-. I could suspect that the population is biased.
  • I would think that number of posterior occlusal support in KC I D- in Table 1 is not 0~4 but 0~3. Would authors perform RPD treatment for the missing of the third molars?
  • Please proof-read the manuscript by a native English speaker.

Author Response

Reviewer 1

Summary of manuscript: This manuscript reports the investigation of the mastication predominance and masticatory performance in KC I patients including the significance of remaining posterior teeth and removable partial denture (RPD) treatment. This report concludes that the difference in the number of remaining teeth between left and right sides might be related to mastication predominance in KC I patients in the absence of RPD because KC I D+ patients showed higher MPI than HD subjects, but not KC I D-.

>Thank you for your comments. The answers and our revisions are described below.

We hope that our revisions can be solved your concerns.

Overall manuscript

  • P2 L61 Authors should describe the difference of number of posterior teeth in KC I, not KC I D+ and D-.

>We hope that you mean the total number of posterior teeth in the subjects because you mention “not KC I D+ and D-“. (KC I D+ means the KC I subjects with the difference of the number of teeth between left and right, as you can find in the body text.) The sentence we wrote in the text, “the significance of remaining posterior teeth, especially the difference of number of teeth, in those has not been evaluated” was used to mean that the total number of posterior teeth and the difference of the number of posterior teeth between left and right in the subjects. It means that we used this description to imply both meanings. Previous studies that evaluated the effect of KC I on MPI was limited. To describe clearly, we revised this sentence.

“However, detailed data of mastication predominance and masticatory function in patients with bilateral missing posterior teeth (Kennedy class I patients; KC I patients) has not been reported, and the significance of remaining posterior teeth, especially the number of posterior teeth in those, has not been evaluated.”

  • Authors should discuss the difference between pre– and post–MPI in KC I D- in Figure 3. If it was the same number of posterior teeth, it would not be no significant differences between pre-and post-MPI in KC I D-. I could suspect that the population is biased.

>This is a very difficult issue as you suggested. MPI in KC I D - without RPD showed no statistical difference compared to HD (Figure 1). However, the treatment of RPD statistically improved MPI in them (Figure 3), although the distribution of MPI in KC I D - was wide range. If we consider that the increase of posterior occlusal support contributes to the improvement of MPI, we need to discuss the effectiveness of RPD treatment in the improvement of MPI. We added the comments the contribution of RPD treatment on MPI improvement to 5th paragraph in Discussion.   

  • I would think that number of posterior occlusal support in KC I D- in Table 1 is not 0~4 but 0~3. Would authors perform RPD treatment for the missing of the third molars?

> Number of posterior occlusal support is defined as total number of posterior occlusal support in both sides, not the difference of number of posterior occlusal support between left and right. The range of posterior occlusal support was from 0 to 6. For example, “4” means that the patients with 1st and 2nd premolar in both sides have 4 posterior occlusal supports. In addition, we’d like to emphasize that the data was described as median and IQR. So this description is correct. The other reviewer recommended to show the distribution of posterior teeth condition, and we added a new table (new Table 2) as you can see in the text.

  • Please proof-read the manuscript by a native English speaker.

>As you suggested, we revised some descriptions after a native speaker’s check.

Reviewer 2 Report

The authors completed a research in a field where they have already published and work consistently. The submission has adequate preparation event hour some minor corrections are needed.

Authors prepared an adequate study on mastication predominance and masticatory performance in Kennedy Class I patients. The study is controlled and prospective. I found the article well described and the results consistent  with the aim of the research. Some minor comment and corrections are needed:

Why the Sample size Is missing . Why the authors evaluated 20 patients per group? Authors should described and justify the number of patients included in this study.

HD paitnets are 25 years old versus a mean age of KC patients 74 years old: this huge difference should be discussed. Why non including a control group with healthy mouth but with similar age. or at least not so young. 

Author Response

Reviewer 2

The authors completed a research in a field where they have already published and work consistently. The submission has adequate preparation event hour some minor corrections are needed.

Authors prepared an adequate study on mastication predominance and masticatory performance in Kennedy Class I patients. The study is controlled and prospective. I found the article well described and the results consistent with the aim of the research. Some minor comment and corrections are needed:

>Thank you for your comments. Our answers and revisions were as described below.

Why the Sample size Is missing. Why the authors evaluated 20 patients per group? Authors should described and justify the number of patients included in this study.

>The subjects were enrolled from only patients the authors treated. During the study period, totally 44 patients were included in this study. We understand that the sample size was not enough to analyze the data, so we added effect size to enhance the validity including new ref #27. The effect size in this study was reasonable and we believe our analyses were meaningful. I hope it will be acceptable for you. In Materials and methods, 2.6. Statistical analysis and in 3rd paragraph in Discussion, the comments were added. In addition, we changed figures and tables into new ones.

HD patients are 25 years old versus a mean age of KC patients 74 years old: this huge difference should be discussed. Why non including a control group with healthy mouth but with similar age. or at least not so young. 

>This is one of the weak points in this study. Actually, in our department, most of the patients were partially edentulous. This means that it was difficult to enroll enough number of HD patients as the subjects in this study. Of course, we need to consider the effect of age on masticatory predominance, but we don’t have data to compare MPI between young and elderly people, and the previous studies also did not evaluate. Unfortunately, this must be research topic in the future. We added the comments regarding this concern in “Materials and methods”, 2.1. Study population and HD group was defined as “positive control”. I hope this revision will be acceptable for you.

Reviewer 3 Report

Please see the comments in the attaeched file.  

Author Response

Reviewer 3

The current work aims at assessing eventual reduced masticatory performance in patients with partial bilateral edentoulism in the posterior area (Kennedy Class I), pre and post- treatment with a conventional removable prosthesis. Also, the study addressed the role of masticatory predominance (using MPI%) based on the consideration that excessive chewing on one side might be risky for the stomatognatic apparatus’ health. Even if the purpose is clear and the clinical experimental design is correct, substantial evidence of the clinical meaning of these parameters is controversial. Therefore, the conclusions should be softened. Please consider some critical points that need revision

: Material and methods

  • Please report more detailed clinical data such as the missing teeth (molars, premolars, superior or inferior). Where the patients homogenous considering the edentulous arch?

>As you pointed, we added a new table (Table 2) in the text.

  • Please group the statistical analyses according to the objectives as reported later in the results section. For ex. comparison of the initial MPI among the groups.

>As you pointed, we added the comments.

Results

  • The number of healthy volunteers is repeated twice, but some data are lacking as for ex. Maximal Occlusal Force (MOF)

>In the previous studies, MOF was the most crucial factor in masticatory performance, as shown in the text and references [22-24]. In this study, the main purpose of measuring masticatory performance was to evaluate the effect of RPD treatment in KC I patients. To demonstrate the validity of the effect of RPD on oral rehabilitation, no difference of masticatory function-related factors (MOF) between KC I D+ and D- was favorable. As a result, we detected no difference of MOF between both groups, suggesting the capacity of KC I subjects were similar except for the number of posterior teeth occlusal support. The role of HD group was a “positive control” in MPI, so we did not measure MOF in HD group. The comments were added in “Materials and methods” and “Results”.

  • Sample size calculation is missing. Was any factor considered in order to recruit about 20 subjects per group? Why healthy subjects were significantly younger than KC I patients? I think that a group of dentate subjects of a comparable age had to be selected.

>As I mentioned as the response to another reviewer, we modified as you can see below.

Regarding the sample size, the subjects were enrolled from only patients the authors treated. During the study period, totally 44 patients were included in this study. We understand that the sample size was not enough to analyze the data, so we added effect size to enhance the validity including new ref #27. The effect size in this study was reasonable and we believe our analyses were meaningful. I hope it will be acceptable for you. In Materials and methods, 2.6. Statistical analysis and in 3rd paragraph in Discussion, the comments were added. In addition, we changed figures and tables into new ones.

Regarding healthy dentate group, this is one of the weak points in this study as you pointed. Actually, in our department, most of the patients were partially edentulous. This means that it was difficult to enroll enough number of HD patients as the subjects in this study. Of course, we need to consider the effect of age on masticatory predominance, but we don’t have data to compare MPI between young and elderly people, and the previous studies also did not evaluate. Unfortunately, this must be research topic in the future. We added the comments regarding this concern in “Materials and methods”, 2.1. Study population and HD group was defined as “positive control”. I hope this revision will be acceptable for you.

  • Is the glucose concentration difference among groups a clinical relevant value? Please report the consideration on the MM and then discuss it.

>As described in the text, this method is widely used to assess masticatory performance, especially in Japan (this method is certified by Ministry of Health, Labour and Welfare of Japan). In this study, masticatory performance is used to assess the effect of RPD on oral rehabilitation. In fact, if we had detected significant difference between KC I D+ and D-, we might have evaluated the association between MPI and masticatory performance. Unfortunately, we did not detect and we’re thinking this can be future research topic.

Some comments were added in “Materials and methods”, 2.5. Objective evaluation of masticatory performance, and “Results”, 3.3. Comparisons of MPI and masticatory performance in patients with KC I (D+ and D-) between pre– and post–RPD treatment.

  • In figures 1 and 2 please add “pre” before KC I D+ and D-.

>As you suggested, we modified.

Round 2

Reviewer 1 Report

The new version of the manuscript was improved and could be considered not to be any problem  for publication.

Reviewer 3 Report

The Authors answered the questions and solved some critical issues by improving the work so that the new version of the manuscript can be considered for publication.